# Prevalence of Compassion Fatigue and Its Association with Professional Identity in Junior College Nursing Interns: A Cross-Sectional Study

**DOI:** 10.3390/ijerph192215206

**Published:** 2022-11-17

**Authors:** Li-Juan Yi, Jian Cai, Li Ma, Hang Lin, Juan Yang, Xu Tian, Maria F. Jiménez-Herrera

**Affiliations:** 1Department of Nursing, Hunan Traditional Chinese Medical College, Zhuzhou 412000, China; 2Department of Nursing, Universitat Rovira i Virgili, 43002 Tarragona, Spain; 3School of Nursing, Yongzhou Vocational Technical College, Yongzhou 425000, China; 4Department of Nursing, Guiyang Medical University, Guiyang 550025, China

**Keywords:** compassion fatigue, professional identity, junior college, nursing intern

## Abstract

*Background:* The issue of compassion fatigue among clinical nurses has received considerable attention, particularly during the COVID-19 pandemic. Yet, the current status of compassion fatigue among junior college nursing interns remains unclear. Additionally, professional identity can modulate the impact of compassion fatigue or burnout on psychological well-being; however, whether professional identity still works in this group is also unclear. This study aimed to reveal the current status of compassion fatigue among nursing interns in junior colleges and also investigate the association between compassion fatigue and professional identity. *Methods*: This cross-sectional survey evaluated the levels of participants’ compassion fatigue (The Compassion Fatigue Short Scale) and professional identity (Professional Identity Scale) in 2256 nursing interns. *Results*: The mean score of compassion fatigue was 44.99, and 19.5% of the participants scored above The Compassion Fatigue Short Scale median scores for compassion fatigue. A moderate negative correlation was detected between compassion fatigue and professional identity. *Conclusions*: The level of compassion fatigue among nursing interns is low but nearly one in five nursing students is at risk of compassion fatigue. More attention should be paid to nursing interns with a high risk of compassion fatigue. Future studies are warranted to explore which pathways could mediate the relationship between professional identify and comparison fatigue.

## 1. Introduction

Nursing is a highly stressful, but compassionate profession. People choosing to be a nurse are empathetic in providing professional help and support according to patients’ needs [1]. However, overexposure to patients’ traumatic experiences during long-term professional engagements could be a source of psychological distress for nursing practitioners, which could erode their ability to take care of patients at an optimum level, as well as their own physical and psychological health [2,3]. This phenomenon was termed as compassion fatigue (CF) by Joinson in 1992, which refers to the formal caregiver’s reduced capacity or interest in being empathic and is considered as the natural consequence of constantly witnessing the traumatizing events suffered by other people [4]. Commonly, CF is composed of secondary traumatic stress (STS) and burnout (BO) [5]. STS is caused by the care provider’s exposure to the suffering of others who have or are experiencing stressful events [6,7], and BO refers to the care provider’s experience of the reduced self-efficacy associated with workload demands and increased perceived stress [6,8,9,10].

The CF of clinical nurses has received wide attention and has been the focus of studies [11]. Evidence shows that the prevalence of moderate-to-severe CF could be up to 57.7%, especially high in clinical nurses who work in the intensive care unit, emergency, oncology, psychiatric and paediatric departments [12]. A recent systematic review involving 28,509 clinical nurses revealed that the levels of CF have increased over time in recent years, and clinical nurses in the Asian regions suffered a severe level of CF symptoms [13]. The issue of CF has become even worse during the COVID-19 pandemic when clinical nurses have been more likely exposed to traumatic and life-altering events [14]. In addition, clinical nurses have to offer more complex care and psychological support for patients suffering from COVID-19 under various pressures, such as enormous nursing workloads, the highest risk of contagion and being unable to bear family responsibilities [15]. The development of CF can weaken clinical nurses’ ability to feel sympathy and empathy at work, therefore hindering the provision of safe, competent, and ethical care [5,16]. Previous studies showed that a large number of nurses were deeply troubled by diverse negative physiological and psychological symptoms related to CF, such as headaches, insomnia, musculoskeletal disorders, depression and professional helplessness [17,18,19,20]. In addition, clinical nurses who suffer from CF can not only manifest low job satisfaction and work engagement, but also show poor job performance and increased medical errors [21,22,23,24]. All of these consequences of CF can induce higher absenteeism and turnover of nursing staff, which are escalating problems faced by healthcare systems [25,26].

Nursing students in junior college are an important reserve of further professional nurses globally. In China, approximately half of all registered nurses held an associate degree until 2019 [27]. Most noteworthy, because nursing students are frequently up against real-life trauma situations and similar workplace environments as professional nurses during their clinical training, they are also vulnerable to CF [28,29,30]. One survey of 972 nursing students found that the prevalence of moderate BO and STS was 97.8% and 55.3%, respectively [31]. However, these studies [31,32,33,34,35] are either specific to undergraduate students or both undergraduate students and junior college students, all of which cannot purely reflect the level of CF among the junior college students. Several studies found that nurses with associate degrees suffered from significantly higher CF than those with bachelor’s degrees [36,37]. One possible reason is that nurses with lower education degrees usually have less rich knowledge reserves. When providing supportive care for patients and dealing with traumatic events in a hospital setting, they may have less professional working ability, and then may achieve less satisfaction from work [38,39]. Given all these, revealing the current status of CF among junior college nursing students is crucial.

Among pre-licensure health profession students, including nursing students, high levels of CF or BO were associated with an increased intention to quit their education program that predicts actual attrition [40,41,42,43,44]. During the COVID-19 pandemic, social distancing and isolation requirements had significantly disrupted healthcare students’ education and clinical rotations, thus students may be unable to develop essential practical skills needed for patient care. These challenges contributed to elevated BO, and uncertainties about the future career, and the increased intention to quit [42,45,46,47,48,49,50,51,52]. In addition, BO developed during the education program could continue to post-graduation and affect new nurses’ well-being and intentions to leave the profession [10]. As a longitudinal study showed that, BO during the internship period was associated with a lower self-rated clinical performance and a higher turnover intentions in one year post-graduation [41]. Obviously, CF or BO will have a negative impact on the career development of nursing students. Hence, understanding how to mitigate CF is of primary importance.

Currently, limited information is available on effective initiatives to alleviate CF. Positive professional identity may be a potential factor that can protect healthcare providers from CF and BO and benefit the maintenance of psychological well-being [12,53,54]. For nursing students, professional identity refers to the process of career planning and confirming their professional role in their current status [55]. A robust professional identity is critical for a nursing student to transform into a confident, successful and resilient healthcare professional [56,57]. Studies shows that professional identity has a positive impact on nursing interns’ response to work stress and caring abilities [58,59].

It is known that a lack of professional identity has detrimental effects on the poor retention of the nursing workforce [60,61,62]. Sabanciogullari et al. [63] found that a professional identity development program significantly decreased clinical nurses’ job BO, suggesting strong professional identity can play a protective role for healthcare providers working at hospitals. Although COVID-19 pandemic poses many risks and challenges associated with healthcare and nursing, it can arouse nursing students’ desire to help others and awareness of the importance of the nursing profession, which are underlying themes of professional identity [45]. A cross-sectional study revealed that the professional identity score among nursing interns increased significantly since the outbreak of COVID-19 [64]. Hence, researchers can utilize the positive impact of responding to the COVID-19 pandemic to cultivate the professional identity of students, thereby alleviating CF. However, only a few researchers have directly investigated the relationship between professional identity and CF among the clinical nurses, but not in the nursing students to date [12,54,65,66]. Therefore, it is necessary to explore the association between professional identity and CF among junior college nursing interns.

This study addressed the following research questions. (a) What is the status of CF among junior college nursing interns in China? (b) Are there significant differences in CF between nursing interns with different sociodemographic characteristics? (c) Is there a relationship between CF and professional identity among junior college nursing interns in China?

## 2. Materials and Methods

We conducted a cross-sectional survey to assess the current status of CF among junior college nursing interns in China and explore the association between CF and professional identity.

### 2.1. Participants

A convenience sampling method was used to select eligible participants from 10 public junior colleges in Hunan Province, China from December 2021 to June 2022. Eligible participants were nursing interns who had completed at least 2 years of nursing-related courses (including nursing students who specialize in general nursing and midwifery), had a clinical internship in a second-level or above hospital, had a clinical internship for no less than 8 months and gave informed consent and voluntary participation in this study. The exclusion criteria were nursing interns whose clinical practice positions were clerical management or administration and not directly in contact with patients.

### 2.2. Sample Size 

The sample size was calculated using the formula:$\;n=\frac{z_{a}^{2}{*\sigma }^{2}}{\delta^{2}}$. There is no study using the CF Short Scale (CFSS) to measure CF among nursing interns. Therefore, the sample size in our study was calculated based on a study reporting the CF among nursing staff in China (standard deviation = 22.88) [67]. Based on the expected error of estimation of 1 and a 5% margin of error, theoretically, a minimal sample size of 2011 was required. A total of 2405 nursing interns were recruited.

### 2.3. Data Collection

The data collection process was carried out through an online survey. The teaching counselors from each college were invited to perform the recruitment. All invited teaching counselors were informed of the research purpose and detailed survey procedures based on standardized trainings to ensure participants meeting eligible criteria and to avoid bias in the data collection process. Sojump (also named as “Wenjuan Star”) software was used to perform the online survey, and the link was sent to participants through WeChat (the most popular and practical instant communication software in China). The average time of the entire answer process is about 15 to 20 min. Completion of the previous question is a mandatory criterion for entering into the next question to ensure the completeness of the questionnaire responses. The purpose and method of this study were explained to the nursing interns who met the inclusion criteria by trained teaching counselors. Each participant read and signed an electronic written informed consent form before filling out the questionnaires. Moreover, they could quit at any time. Questionnaires with identical answers or regular answers were excluded. After removing 149 invalid responses, we were left with a final count of 2256 (an effective response rate is 93.8%).

### 2.4. Measures

#### 2.4.1. Demographic Characteristics

The demographic characteristics questionnaire was designed by the research team to collect participants’ gender, age, specialty, duration of an associate program, whether being from an only child family, frequency of night shifts, home location (rural or urban), monthly living expenses and employment intentions.

#### 2.4.2. CF Short Scale (CFSS)

Nursing interns’ CF was measured by the CF Short Scale. The original CFSS was developed and tested by Adams et al. (2006) [68], consisting of 13 items in 2 dimensions including STS (five items) and BO (eight items). The Cronbach’s alpha coefficient of the total scale, STS subscale and BO subscale was 0.90, 0.80 and 0.90, respectively. Each item is scored at a 10-point Likert scale, ranging from 1(never) to 10 (always), with a total score of 13–130. The higher the score indicated the more severe CF. The present study used the Chinese version of the CFSS, which was translated and validated by Sun (2015) [69], to measure CF. In this study, the Cronbach’s α of the total scale, STS subscale and BO subscale was 0.917, 0.857 and 0.888, respectively.

#### 2.4.3. Professional Identity

Professional identity was measured using the Professional Identity Scale which was developed by Brown et al. [70]. The Professional Identity Scale is a unidimensional scale and consists of 10 items. Of these 10 items, five are positive and the other five items are negative. Items are rated on a 5-point rating scale from 1 (never) to 5 (always), with a total score of 10–50. The higher score indicated the stronger professional identity. The Cronbach’s coefficient of the original scale is 0.71. This study used the Chinese version of this scale, which has been translated and validated by Lu et al. [71] with a Cronbach’s coefficient of 0.82, to measure professional identity. The Cronbach’s of the scale in this study was 0.803.

### 2.5. Data Analysis

Data analysis was conducted using SPSS (version 25.0) software (IBM Corp., Armonk, NY, USA). A two-tailed *p*-value less than 0.05 was deemed statistically significant. First, participants’ baseline characteristics were summarized as frequency, percentage, mean and standard deviation (SD). We further classified the score of CFSS into dichotomous variables by using the median of the total score as the cut-off value [72]. Specifically, if participants who reported a total score of CFSS ≥ 65, BO subscale ≥ 40 or secondary trauma subscale ≥ 25, they were considered to be at high risk for CF, BO and STS. Secondly, we used the Kolmogorov–Smirnov test to evaluate the distribution of the total score of the CFSS and Professional Identity Scale, suggesting abnormal distributions of the two scores (CF: D = 0.092, *p* < 0.001; professional identity: D = 0.079, *p* < 0.001). However, according to the following two reasons, we selected the parametric test to compare the difference of CF between participants with different categories: (1) when the data sample size is big enough, the normality test result is unstable and not always reliable [73] and (2) the central limit theorem also confirmed that the sampling distribution tends to be normal in large samples, despite the shape of the data [74,75]. Therefore, independent *t*-test and one-way ANOVA was used to assess the difference of CF level between participants from different groups. The significant factors were then included in the linear regression to further test which baseline factors could contribute to the development of CF. The enter strategy was used when conducting the multivariate linear regression analysis. Pearson correlation was used to explore the association between CF and professional identity. Cohen’s rule of thumb has become a de factor standard for this effect size [76], suggesting that r = 0.10 was “small”, r = 0.30 was “medium” and r = 0.50 was “large”. However, we also performed the non-parametric analysis to examine robustness of the findings. 

## 3. Results

### 3.1. Participants’ Socio-Demographic Characteristics

Participants’ characteristics are presented in Table 1. The majority of participants who returned the validated questionnaires were female (n = 2077, 92.1%) and from the nursing specialized field (n = 1660, 73.6%). Most of the participants have undertaken a three-year nursing course (n = 1736, 77%) and have their internship in the tertiary hospital (n = 1708, 75.7%). The majority were from the urban areas (n = 1784, 79.1%), not the only child in their family (n = 1962, 87%) and had monthly expenses between 1000 to 2000 yuan (69.8%). About half of the participants have the experience of being student leaders (n = 1172, 52%) and 86.1% of nursing interns intend to be a nurse or midwifery. 

Significant differences were found among five factors from the results of the independent *t*-test or one-way ANOVA test (Table 1). Specifically, participants who specialized in midwifery were female, had internships at secondary hospitals and intended to be a nurse or midwife after graduation reported significantly higher CF levels than those who specialized in general nursing (mean difference [MD]: −2.89, 95% confidence interval [CI] −4.91 to −0.83), were male (MD: −4.93, 95% CI: −8.37 to −1.48), had an internship at a tertiary hospital (MD: −4.26, 95% CI: −6.48 to −2.04) and did not intend to become a nurse and midwifery after graduation (MD: −15.55, 95% CI −18.51 to −12.59). Additionally, participants who experienced more night shifts reported higher levels of CF than those who experienced relatively fewer night shifts.

### 3.2. The Status of CF

The mean and SD of the score of total CF was 44.99 ± 22.58, with 29.26 ± 15.04 for BO and 15.73 ± 9.41 for STS (Table 2), all of which were considered a positive indication of weak CF among nursing interns. Median values were used as a threshold to divide the total score of CFSS and the two CFSS subscales into two levels. There were 440 (19.5%) participants reporting an overall CF score above 65 and were regarded as having high levels of CF. In addition, 24.2% of the respondents had BO and 17.2% showed symptoms of STS.

### 3.3. Multiple Linear Regression Analysis of Influencing Factors

Table 3 displays the results from the multiple linear regression analysis. Five factors (specialty, gender, level of hospital, the number of night shift per month and employment intention) with a *p*-value of <0.05, showing a statistically significant association with a *t*-test or ANOVA, were placed in the multiple linear regression model. The tolerance values ranged from 0.958 to 0.989, indicating no multi-collinearity. The results of linear regression demonstrated a positive association among four personal factors (midwifery, secondary hospital, six night-shifts per month and no intent to be a nurse or midwife) and CF. However, these four variables only explained 6.8% of the total variance CF (*p* < 0.001). It indicates that there are other factors besides demographic and sociological factors and at the level of the inter hospital, there are other factors that should be considered which could explain the CF level of nurse interns 

### 3.4. The Correlation between CF and Professional Identity

The mean and SD of the Professional Identity Scale among nursing interns was 38.45 ± 6.18. The results of correlation analysis identified a weak negative correlation between professional identity and CF (r = −0.42, *p* < 0.001), STS (r = −0.25, *p* < 0.001), BO (r = −0.47, *p* < 0.001) (Table 4). Therefore, this is indicative of a significant relationship between these variables and suggests that as nursing students suffer from CF, the likelihood of obtaining a low level of professional identity is minimal during their internship period.

## 4. Discussion 

### 4.1. Main Findings 

We performed this cross-sectional study to (1) explore the prevalence of CF among Chinese nursing interns in the associate program; (2) compare the CF level among nursing interns from different sociodemographic and institution characteristics; (3) find out the relationship between nursing interns’ CF and their professional identity. The results of our study suggested the average level of CF among nursing interns is low, but about one in every five students is at high risk for CF and more participants have BO than STS. Nursing interns who specialized in midwifery reported higher CF than those who specialized in nursing. Nursing students who had an internship in a secondary hospital showed higher CF than those who had an internship in a tertiary hospital. We also found that the more night shifts they undertook, the higher level of CF they reported. Overall, 13.9% of participants who reported that they intended to withdraw from the nursing or midwifery profession, reported much higher CF than those who intended to stay in this profession. In addition, correlation analysis identified a significant medium association between CF and professional identity.

### 4.2. The Status of CF

Although CF among clinical nurses has received widespread attention, there is limited published research focusing on nursing students [32,77]. This study found that 19.5%, 24.5% and 17.2% of participants presented with a high risk for CF, BO and STS by using the median as cut-off value, respectively, which indicate that the CF level among nursing interns is low. This result was in line with the findings of Han Binru et al. (2017) who found that 37.8% and 21.9% of nursing interns reported high levels of BO and STS, respectively [78]. Likewise, Cao el al (2021) reported very low prevalence in high levels of BO and STS among nurse interns (0.9% and 1.1%, respectively) [32]. Though these studies used different measurement tools and classification methods, they all reported a low-level CF among nursing interns

CF is caused by continual exposure to stress and trauma during clinical practice. Therefore, the CF level among healthcare providers is associated with the time and nature of their work. The low level of CF among nursing interns is a possible reason as nursing students receive only 8 to 10 months of clinical internship in different departments. Another reason for the low level of CF among nursing interns could be that nursing students are protected from medical errors under the strict supervision of clinical teachers, which could mitigate their stress from work [79]. However, nearly 20% of participants who had a high level of CF still merited attention by educators and clinical managers. The clinical internship provides students with opportunities to practice their knowledge and skills learned in classrooms. However, this process would be stressful for some nursing students as they lack clinical knowledge and self-confidence. Nursing interns could be at the bottom of the social hierarchies in the stressful clinical settings. Studies report that nursing interns are at risk of experiencing bullying from staff nurses, clinical instructors, as well as patients and families, which could be the reason for BO, high CF and low career identity. Further studies are needed to explore the reasons for the high level of CF among these participants. Interventions should be developed to assist them in building their competencies and confidence, the actualization of their role as nurses, and the understanding of their responsibilities in care, which could enhance their enthusiasm for helping others [35].

### 4.3. Comparison of CF by Participants’ Characteristics

When we investigated the distribution of CF among the participants, we found female participants, those from a midwifery speciality, those with internships at secondary hospitals and who had more night shifts as well as those who intended to switch to another profession after graduation were associated with a higher level of CF. 

Regarding area of speciality, participants whose specialty was in midwifery showed higher CF than for those with a specialty in nursing. During the clinical internship, both nursing and midwifery students face the challenge of demanding workloads, challenging placements and witnessing traumatic events, with subsequent stress sometimes affecting CF. Compared with nursing students, students who specialize in midwifery are more likely to work in the units related to birth, which are considered happy events that bring positive emotions. However, there are factors such as loss of control, inability to cope with pain, various complications and traumatic birth that can turn the childbirth process into a stressful and traumatic experience [80,81]. A recent study showed that midwifery students were facing a high level of STS rates which should be supported by mental health nurses to cope with traumatic stress [82]. However, further studies should explore the difference between midwifery and nursing interns.

Regarding level of hospital internship, nursing interns in the secondary-level hospital had higher CF than nursing interns in the tertiary hospital. The classification of hospitals in China is a three-tier system: primary, secondary and tertiary hospitals. Secondary hospitals are responsible for providing comprehensive health services and managing simple diseases. Tertiary hospitals provide specialist health services and serve as medical hubs providing care to multiple regions. Reports show that compared with tertiary-level hospitals, the secondary-level hospitals face the greater challenges of staff shortages, are less advanced relative to the staff’s level of education and have less sophisticated medical equipment. These challenges lead to higher workloads and a higher risk of BO for healthcare workers in secondary-level hospitals and also hurt the quality of medical outcomes for patients. As for nursing staff, nursing interns will experience the same stress in secondary-level hospitals which will increase their CF and BO [83].

In the aspect of number of night shifts, nursing interns who undertook more night shifts reported higher scores of CF. This finding is consistent with early studies [84,85]. As for nursing staff, nursing interns also need to take on different work shifts. Evidence shows that shift work can cause BO for the nurses as well as a series of health problems (Hughes 2015; Wang 2019). Shift work also has a negative impact on nurses’ job satisfaction and job performance [86,87]. In addition, nursing students in the post-internship stage are faced with the pressure of job hunting and examinations, and they are more concerned about the impact of scheduling. Research shows that about 63% of nursing students report decreased work enthusiasm and job satisfaction from shift work, which also has a negative impact on their communication with patients [78]. Therefore, managers should appropriately refer to the opinions of nursing students, arrange flexible shifts, improve the practice efficiency of nursing students, and reduce the negative pressure of scheduling on nursing students.

In the aspect of intent to be a nurse or midwife, there are about 14% of participants reporting that they intend to withdraw from the nursing profession [88,89]. These participants have significantly higher CF than those who would like to stay in this profession. Clinical internships provide nursing students with chances to practice the knowledge learn from the classroom and also provide them with a chance to gain an insight into the real nurses’ roles and meanwhile expanding the expectations of their future careers. A qualitative interview shows that two reasons for nursing students’ dropout in the late stage of their study are lacking a safe learning environment in clinical placements and psychological support, and realizing that the training and the future profession did not match their expectations and wishes [90]. CF could be a reason for nursing intern intent to withdraw from the nursing profession. Conversely, nursing interns who intend to show less enthusiasm for their work and patients, as a result, are more exposed to negative emotions, including CF. This finding is further tested by the negative correlation between CF and professional identity.

### 4.4. The Association between CF and Professional Identity

This study found a negative correlation between CF (including both STS and BO) and professional identity. The negative association between CF and professional identity has also been tested among general nurses [54], operation room nurses [66] and ICU nurses [12]. Professional identity was described as a person’s perception of themselves within a profession or the collective identity of the profession, which plays an important role when nursing students decide to choose their career. The formation of a professional identity is an evolving process, shaped by educational experiences, life experiences, work experiences and social media. Clinical internships provide nursing students with a chance to explore and experience real clinical nursing work CF, including both BO and STS, is a resource of work-related stress. It has a negative impact on nursing interns’ well-being and then further reduces their passion for the nursing profession. On the one hand, the way nursing interns negotiate their professional identities could soothe or exacerbate the impact of CF on their mental health. A high level of professional identity can improve healthcare workers’ psychological state and make them more resistant to CF [91].

### 4.5. Limitation and Further Studies

There are several limitations in the present study. First, the cross-sectional study design and self-report data as opposed to objective measurement, and therefore our results may be negatively influenced by potential bias. Second, convenience sampling was used in this study, which might be a source of selection bias. Third, although a significant difference was reported among participants with different baseline characteristics, these factors can only explain 6.8% of the CF. Therefore, further studies are needed to explore the predictors of CF. Moreover, further studies could use qualitative methods to explore the reasons for those nursing interns who reported high CF. 

## 5. Conclusions

The study found that the CF level among Chinese nursing students is lower than that among the nurse staff. Participants with a midwifery speciality, having internships at secondary hospitals and had more night shifts as well as those who did not intend to work as a nurse or midwife reported a high level of CF. There is a negative relationship between CF and nursing interns’ professional identity. Future studies are needed to explore which pathways could mediate the relationship between professional identify and comparison fatigue among nursing interns.

## Figures and Tables

**Table 1 ijerph-19-15206-t001:** The score of CF according to various characteristics.

Variable	Categories	N (%)	Average Score of CF		
			Mean ± SD	t/F	*p*-Value
Specialty	Nursing	1660 (73.6%)	44.23 ± 22.97	−2.671 *	0.008
	Midwifery	596 (26.4%)	47.11 ± 21.35		
Degree	Three years course	1736 (77.0%)	44.9 ± 22.65	−0.374 *	0.709
	Five years course	520 (23.0%)	45.32 ± 22.39		
Gender	Male	179 (7.9%)	40.46 ± 22.45	−2.804 *	0.005
	Female	2077 (92.1%)	45.38 ± 22.56		
The place of birth	Urban area	472 (20.9%)	44.02 ± 21.91	−1.051 *	0.293
	Rural area	1784 (79.1%)	45.25 ± 22.76		
Only child	Yes	294 (13.0%)	43.68 ± 22.85	−1.066 *	0.287
	No	1962 (87.0%)	45.19 ± 22.54		
Monthly expense	<1000 yuan	420 (18.6%)	44.50 ± 21.39	0.248 ^#^	0.780
	1000–2000 yuan	1574 (69.8%)	45.00 ± 22.94		
	>2000 yuan	262 (11.6%)	45.76 ± 22.32		
Experience of being student leaders	Yes	1172 (52.0%)	45.15 ± 22.92	0.352 *	0.725
	No	1084 (48.0%)	44.82 ± 22.22		
Level of intern hospital	Tertiary hospital	1708 (75.7%)	43.96 ± 22.24	−3.858 *	<0.01
	Secondary hospital	548 (24.3%)	48.22 ± 23.34		
No. of night shifts (monthly)	0–2/month	803 (35.6%)	43.67 ± 21.56	5.471 ^#^	0.001
	2–4/month	1037 (46.0%)	44.59 ± 22.49		
	4–6/month	278 (12.3%)	47.27 ± 22.84		
	>6/month	138 (6.1%)	51.19 ± 27.10		
Intent to be a nurse or midwifery	Yes	1943 (86.1%)	42.83 ± 21.36	−11.642 *	<0.01
	No	313 (13.9%)	58.39 ± 25.24		

Note: CF: compassion fatigue; SD: standard deviation; IQR: interquartile range; *: *t*-test statistic; ^#^: one-way ANOVA statistic (F statistic).

**Table 2 ijerph-19-15206-t002:** Distribution of responses to CFSS and its subscales, and median score among nursing interns (n = 2256).

Variables	Mean ± SD	Distribution of Responses (%) ^a^	
		Low (N, %)	High (N, %)
CF	44.99 ± 22.58	1816 (80.5%)	440 (19.5%)
BO	29.26 ± 15.04	1711 (75.8%)	545 (24.2%)
STS	15.73 ± 9.41	1867 (82.8%)	389 (17.2%)

Note: CF: compassion fatigue; burnout: BO; STS: secondary traumatic stress; CFSS: the CF Short Scale; IQR: interquartile range; SD: standard deviation; ^a^: preset median value as a cut point to classify the total score of CFSS and two subscales into two levels, respectively. If participants with a total score of CFSS greater than or equal to 65, they were considered to be at high risk for CF. If participants had a total score of BO subscale greater than or equal to 40, they were considered to be at high risk for BO, and if participants had a total score of STS subscale greater than or equal to 25, they were considered to be at high risk for STS subscale.

**Table 3 ijerph-19-15206-t003:** Multivariate linear regression analysis of CF among medical interns (n = 2256).

Variable	Categories	*B*	SE	*Beta*	*t*	*p*-Value
Specialty	Nursing (reference)					
	Midwifery	2.573	1.063	0.05	2.421	0.016
Gender	Male (reference)					
	Female	2.394	1.735	0.029	1.379	0.168
Level of hospitals	Tertiary hospital (reference)					
	Secondary hospital	3.902	1.076	0.074	3.628	<0.01
No. of night shifts per month	2–4/ month (reference)					
	0–2/month	−0.628	1.027	−0.013	−0.612	0.54
	4–6/month	1.393	1.478	0.02	0.943	0.346
	>6/month	1.510	0.549	0.056	2.753	0.006
Intent to be a nurse or midwifery	Yes (reference)					
	No	15.047	1.335	0.230	11.270	<0.01
Constant term		12.304	3.863		3.185	0.001

Adjusted R² = 0.068; F = 33.794; *p*-value < 0.000.

**Table 4 ijerph-19-15206-t004:** The correlation between CF and its two domains and professional identity.

Variables	CF	STS	BO	Professional Identity
CF	1			
STS	0.88 **	1		
BO	0.95 **	0.69 **	1	
Professional identity	−0.44 **	−0.26 **	−0.49 **	1

Note: CF: compassion fatigue; STS: secondary traumatic stress; burnout: BO; **: *p* < 0.001 for two-tailed.

## Data Availability

Upon reasonable request, the original data supporting the conclusions of this article can be accessed from the corresponding authors.

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
