# Peer review of "Prevalence of Compassion Fatigue and Its Association with Professional Identity in Junior College Nursing Interns: A Cross-Sectional Study"

_ijerph, 2022, doi:10.3390/ijerph192215206_

Round 1

Reviewer 1 Report

The present study is of great significance as healthcare professionals are regularly exposed to difficult and stressful situations in their line of work. However, discussing burnout, especially during critical times such as the Covid-19 pandemic, should consider recent studies on this particular topic.

Therefore, I kindly suggest updating the references as there are several studies on medical students' and nurses' burnout.

My improvement suggestions are the following:

27 - elevated professional identity - is ”elevated” the proper term?

The introduction mentions the term ”burnout”: is burnout correctly conceptualized and presented through a body of literature on the topic? The article could note how burnout is defined, as recent studies show that is a multidimensional concept. This is also related to the negative correlation between professional identity and CF, which eventually is shown to have a protective effect on CF, and, subsequently, on burnout dimensions. 

One mention on the thousands separator: the text has a comma and no separator at all, it should be corrected and unified (line 45).

Lines 54-55 - maybe replace the term ”deleterious” and rearrange the psychological impact and physiological symptoms, as they are not on the same conceptual level. 

Line 69 - please rephrase.

On Material and Method please improve the paragraph; it should mention how the participants had only one time opportunity to fill in the questionnaire and how the researchers assured that all participants were nurses and they only filled out one questionnaire per person.

Line 140 - maybe rephrase the process which does not allow respondents to skip questions.

Also, in Data Collection chapter please see the verbs' tense. It might be necessary to switch to the past tense.

Line 170 - I suggest citation be placed immediately after the authors' mention (Lu et al.)

Line 190 - is Spearman to be italicized? Maybe regular font is the one to be used. 

Line 207 - there is an "s" missing at Wallis

Line 358 - there should be a pause before the citation after "nurses"

As for the conclusions, it would be interesting to see again how a multidimensional concept as CF is moderated by professional variables, which, eventually impacts the recommended interventions to reduce CF. 

Reviewer 2 Report

I read the paper entitled "Prevalence of compassion fatigue and its association with professional identity in junior college nursing interns: a cross-sectional study".  I should emphasize that the strength of the study is the sample size. On the other hand, the poor level of statistical processing of the sample can be criticized.

Here are my comments:

 Abstract

- Abbreviations in the abstract would be better removed.

 Introduction

-Line 48 correct reference (Arnetz 2020).

 Data analysis

- To check the normality of the distributions, it would be more understandable for the readers if researchers conduct the Shapiro Wilk or Kolmogorov Smirnov test.

I should add that due to the large sample size of the study, many researchers would argue for the use of means, invoking the Central Limit Theorem.

 Table 2

For the above reason I would suggest adding means and standard deviations to the table.

 Still to improve the statistical results, I would suggest using a regression method.

 Discussion

-Correct the references on lines 297-303.
